# Features of Profiles of Biologically Active Compounds of Primary and Secondary Metabolism of Lines from VIR Flax Genetic Collection, Contrasting in Size and Color of Seeds

**DOI:** 10.3390/plants11060750

**Published:** 2022-03-11

**Authors:** Elizaveta A. Porokhovinova, Tatyana V. Shelenga, Yulia A. Kerv, Valentina I. Khoreva, Alexey V. Konarev, Tamara V. Yakusheva, Andrey V. Pavlov, Anastasia A. Slobodkina, Nina B. Brutch

**Affiliations:** N. I. Vavilov All-Russian Institute of Plant Genetic Resources (VIR), B. Morskaya Str. 42–44, 19000 St. Petersburg, Russia; e.porohovinova@mail.ru (E.A.P.); t.shelenga@vir.nw.ru (T.V.S.); kerv@mail.ru (Y.A.K.); horeva43@mail.ru (V.I.K.); a.konarev@vir.nw.ru (A.V.K.); yakusheva.vir@yandex.ru (T.V.Y.); avpavlov77@yandex.ru (A.V.P.); snas1999@yandex.ru (A.A.S.)

**Keywords:** *Linum usitatissimum*, metabolomic profiling, biochemical composition of seeds, genetic collection, correlations

## Abstract

Flax is one of the oldest oil crops, but only since the end of the twentieth century nutritional use of its whole seeds and flour has been resumed. This crop has been evaluated for its oil fatty acid composition, content of sterols and tocopherols, carbohydrate composition of mucilage, but a comprehensive study has never been carried out, so the aim of the work was to identify differences in the metabolomic profiles of flax lines contrasting in color and size of seeds. The biochemical composition of seeds from 16 lines of the sixth generation of inbreeding was tested using gas chromatography coupled with mass spectrometry. In total, more than 90 compounds related to sugars (78% of the identified substances), free fatty acids (13%), polyatomic alcohols (5%), heterocyclic compounds, free amino acids, phytosterols and organic acids (no more than 2.5% in total) were identified. Statistical analyses revealed six main factors. The first is a factor of sugar content; the second one affects most of organic acids, as well as some free fatty acids, not related to reserve ones, the third factor is related to compounds that play a certain role in the formation of “storage” substances and resistance to stress, the fourth factor is influencing free polar amino acids, some organic and free fatty acids, the fifth one is a factor of phenolic compounds, the sixth factor combined substances not included in the first five groups. Factor analysis made it possible to differentiate all 16 lines, 10 of which occupied a separate position by one or two factors. Interestingly, the first two factors with the highest loads (20 and 15% of the total variability, respectively) showed a separate position of the gc-432 line, which differed from the others, not only by chemical composition, but also by the phenotype of the seeds, while gc-159 differed from the rest ones by the complex of organic acids and other substances taking about 1% of the extracted substances of the seed. Thus, the analysis of metabolomic profiles is promising for a comprehensive assessment of the VIR flax genetic collection, which has wide biochemical diversity.

## 1. Introduction

The Metabolomics and metabolomic profiling is a relatively young branch of biochemistry. This term was proposed by S. Oliver in 1998 [1]. From the genetic point of view, the metabolic profile can be considered as a biochemical realization of the genotype [2]. The methodological basis of this approach is chromatographic analysis in combination with mass spectrometry [3,4]. It allows a comprehensive assessment of the biochemical composition of a particular object of evaluation, as well as the processes occurring in living objects. In recent years, metabolomic profiling has been widely used in biology and agricultural science: phenotyping of species and cultivars, genetic modification, analysis of resistance to biotic and abiotic environmental factors [5,6].

Practically all substances included in the metabolomic profile can be attributed to biologically active ones [7], i.e., substances having physiological activity at low concentrations.

Often the same substances act at the same time as protective agents against plant pathogens, medicine, flavoring and anti-nutritional matters. Depending on the concentration, they can be both useful and harmful [8], for example cyanides. Therefore, it is important for the food and pharmaceutical industry to determine the exact content of the target substances in the plant material. Metabolomic profiling, both based on gas and liquid chromatography with mass spectrometry, allows carrying out a comprehensive assessment of the biodiversity of accessions from world collections of plant genetic resources, including the N.I. Vavilov All-Russian Institute of Plant Genetic Resources (VIR) collection.

In VIR, metabolomic profiling is used to identify differences between species [9,10], subspecies [4], cultivars [4,11,12], forms resistant and susceptible to pathogens [13,14].

The three main cereal crops (wheat, rice, maize) account for 60% of the world’s food crop production and do not satisfy humanity’s needs for proteins and other nutrients. Therefore, FAO recommends diversifying the human diet with previously “forgotten products” that are more optimal in processing, useful for health, while at the same time are economically supporting the local population, improving agroecology and developing regenerative agriculture [15].

One of such crops is linseed, the cultivation of which for seeds production, both for nutrition and for medical purposes, began 5000 BC [16]. This crop is used for functional nutrition as a rich source of α-linolenic acid, lignans, high-quality protein and soluble dietary fiber [15]. Flax seeds contain on average 41% fat, 20% protein, 28% dietary fiber, 7.7% water and 3.4% ash [17]. Flax proteins are rich in arginine, aspartic and glutamic acids, opposite lysine, methionine and cysteine are limiting [18].

It should be noted that until now, linseed is mainly cultivated for industrial oil-for the production of drying oils, varnishes, paints, inks, cosmetics [19], while cake and meal are a by-products and are mainly used as a highly nutritious additive to feed in animal husbandry [20]. Since the end of the twentieth century, flax has been widely used in nutrition as a “whole-grain” additive to dough in baking, and then as flour (whole-grain or fat-free) both in combination with wheat flour and independently [21,22]. Now, in addition to bakery products, flax seeds are included in dairy products, as well as pasta. In the latter, flax flour 15% *w*/*w* increases the shelf life of pasta due to its antimicrobial activity. The addition of 10% milled linseed increases the volume of the loaf and slows down its staling [23].

Flax seeds are actively used in the pharmaceutical industry. Oil rich in α-linolenic acid is a very popular commercial product due to its anti-atherogenic, anti-inflammatory and anti-proliferative effects and is used in the complex therapy of obesity. Currently, linseed oil is introduced into the composition of medicines, for example nanoemulsions, to improve their digestibility [15].

Flax proteins and lignans together with the mucilage complex are traditionally used for the treatment of gastritis, lowering blood cholesterol, treating atherosclerosis, diabetes, nephritis and hormone-dependent cancer [15,22,24,25,26]. Flax proteins can serve as a source of biologically active peptides with antihypertensive or immunosuppressive effects [15].

Mucilages, as well as oils, are used for carrying of drugs, but in contrast with the latter ones, they provide a prolonged effect of drugs on target organs in cancer chemotherapy and iron deficiency anemia. Rhamnogalacturonan of flax mucilage causes its prebiotic activity. A mixture of peptidoglycans and polysaccharides has already been included in a commercial product that reduces skin sensitivity in inflammation, another polysaccharide product is recommended as a saliva substitute for xerostomy [15].

Now another strategy of flax seeds processing has begun spread. At the first stage, the coat rich in polysaccharides is separated from cotyledons, which are rich in fat. Then these fractions are processed separately to obtain pure polysaccharides, oil and protein [15].

Linseed has a relatively low variability in seed color. Yellow-seed forms are mainly used in baking [23], however, seeds of other colors are in demand for the production of “multigrain” bread [21].

Flax seeds are quite well evaluated for their fatty acids composition [27,28], polysaccharides [26] and mineral composition [29,30], proteins content, their amino acid composition [17,29], cyanogenic glycosides [30,31], phenolic compounds [25,32,33,34], including lignans and phytin [29,35], however, we have not found any investigations on the comprehensive evaluation of seeds. Traditional biochemical analyses characterize only separate groups of substances. They can’t show comparative amounts of different groups of substances. So, this method makes it possible to compare the amounts of very different substances and reveal the intraspecific diversity of plants biochemical composition and interrelationships between groups and individual substances. Till now practically no investigations of flax metabolome were carried out. That’s why the main goal of our research was to reveal flax intraspecific diversity of seeds biochemical composition using genotypes which represent wide scale of intraspecific diversity.

In order to give a comprehensive assessment of the crop as a source of economically valuable substances, including biologically active ones, it is necessary to fulfill not only a comprehensive evaluation of several typical representatives, but also the identification of biological diversity of these compounds.

A comprehensive study of the intraspecific diversity of cultivated plants stipulates a careful selection of research objects and methods. World collections of plant genetic resources and genetic collections, created on their basis are both the best object and a tool for solving this problem [36].

The purpose of this work was the detection of differences in metabolic profiles, including biologically active compounds of linseed lines, contrasting in their color and size.

## 2. Results and Discussion

### 2.1. Main Groups of Substances in Flax Seeds

For the convenience of analysis, the identified substances are divided into six groups. The group of sugars included mono-, di- and tri-sugars, as well as glyceraldehyde, the group of fatty acids consisted of free fatty acids, mono- and diacylglycerols, as well as paraffin. The group of amino acids included carbamide, in addition to free amino acids. Secondary metabolites meant phytosterols, phenolic, and heterocyclic aromatic compounds. Separate groups were formed by polyatomic alcohols, including glycerol-3 phosphate and organic acids, including phosphoric acid (Table 1).

Main group of the isolated substances was formed by sugars (78%), followed by free fatty acids (13%) and polyatomic alcohols (5%). Secondary metabolites, free amino acids, phytosterols and organic acids were detected in minor quantities, no more than 2.5% each (Figure 1, Table 1 and Appendix A).

Line gc-432 with black seeds has a significantly higher amount of total identified substances in comparison with the others (13,802 n.u.), the smallest amount have gc-109 and 391 (5983 n.u.). Gc-432 is the leader in the content of sugars, secondary metabolites, free amino acids and organic acids, gc-136 is the leader in the content of free fatty acids, secondary metabolites and polyatomic alcohols, gc-173 is the best for free amino acids. Gc-109–has the smallest amount of all free fatty acids, polyatomic alcohols, secondary metabolites, free amino acids and organic acids, gc-391 has minimum sugars and free amino acids, gc-119, 121, 210 form very little polyatomic alcohols, gc-129 is lack of organic acids (Appendix A).

Correlation analysis showed that these substances form two pleiades. The first one includes organic acids, free amino acids and sugars, the second–free fatty acids, polyatomic alcohols and secondary metabolites. The pleiades are interconnected due to a moderate correlation between secondary metabolites and free amino acids (Figure 2a, Appendix A).

The obtained data differ greatly from the results of classical biochemical analysis, according to which the fatty acids of the oil make up about 43%, proteins 10–33%, and sugars about 26% of the seed weight [24,29], which is obviously due to different methodological approaches. So, with the help of the method we used, sugars are extracted much easier than fatty acids, and proteins are practically not isolated.

On the other hand, the results obtained using standard techniques also differ. When comparing the fat content of brown high-linolenic seeds and yellow low-linolenic ones, it was shown that they practically do not differ from each other (45.2 and 44.0 g/100 g of seeds, respectively). However, the brown-seeded form has more oil left in the cake than the yellow-seeded one (12.1 and 7.55 g/100 g of cake, respectively). Thus, the total oil content was very different (52.0 and 48.3 g/100 g of seeds). In the same experiment, using standard methods of proteins and carbohydrates extraction by acid and alkali, it was found that: the fraction of acid solution protein of brown seeds contained only about 66% of protein, and that of yellow seeds contained 23% of protein; the fractions of acid solution carbohydrate contained 55 and 75% carbohydrates, respectively. Thus, the results of the analysis depended both on the features of the extraction method and on the genotypes of the analyzed accessions [37,38].

### 2.2. Sugars Composition of Flax Seeds

In our experiment, the isolated sugars are mainly represented by oligosaccharides (in total >95%): sucrose (**Suc**)-3262 n.u., raffinose (**Raf**)-3080 n.u.) and melibiose (**Melib**)-981 n.u. More than 0.4% accounted for mannose (**Man**), arabinose (**Ara**), glucose (**Glc**), turanose glucopyranoside (**Tur**), fructose (**Fruct**), ribose (**Rib**) and maltose (**Malt**)-60, 55, 54, 41, 38 n.u., respectively, and less than 0.3%-for xylose (**Xyl**), galactose (**Gal**), sorbose (**Sorb**), 2-O-glycerol-d-galactopyranoside (**2OgliDgal**), methylglucoside (**methGluD**), glyceraldehyde (**glAld**) and lyxose (**Lyx**) 0.4, 0.4, 0.23, 0.14, 0.13, 0.12, 0.04, 0.03, 0.02 n.u., respectively (Figure 3a,b, Table 1 and Appendix A).

Significantly more sugars in comparison with other lines, are formed by the black seeds line gc-432 (**Xyl, Ara, Rib, Fruct, Sorb, Gal, Glc, Suc, Raf, Tur, 2OgliDgal**), as well as gc-119 (**Lyx, methGluD, Malt, Raf, Tur**), gc-141 (**Glc, Melib**), gc-121 (**Melib, Malt**), gc-210 (Glc), gc-173 (**Man**), gc-473 (**methGLuD**), gc-2 (**Suc**), gc-136 (**Malt**), gc-159 (**glAld**). The yellow-seeded solin line gc-391 has significantly less of the following sugars in comparison with other genotypes: gc-391 (**Xyl, Ara, Sorb, Gal, Glc, Suc, Raf, Tur**,) and gc-65–(**Lyx, Rib, Gal, methGluD, Melib, Malt, 2OgliDgal, glAld**), gc-2 (**Lyx, Man, 2OgliDgal, glAld**), gc-132 (**Fruct, Sorb, Glc, methGluD**), gc-136 (**Fruct, Sorb, Gal, Glc**), gc-159 (**Xyl, Ara, Rib**), gc-109 (**Lyx, methGluD, 2OgliDgal**), gc-210 (**Ara, Rib**), gc-432 (**Man, Malt**), gc-124 (**Malt**) (Figure 3b, Appendix A).

Correlation analysis of these characters showed the presence of a large pleiad, including closely connected **Xyl, Ara, Fruct, Sorb, 2OgliDgal, Tur, Rib** and **Gal**, as well as adjacent **Glc, Suc** and **Raf**. The second pleiad of characters, weaker related to each other, was formed by **Lyx**, correlated with **Malt** and **Melib**, the latter was associated with **Glc** and **Raf** from the first pleiad. **MethGluD** had a moderate correlation with **Tur**. Interestingly, all significant correlations are positive. In addition to that, their concentrations do not depend on the amount of the precursor of the biosynthesis of all sugars–glyceraldehyde (Appendix A). On one hand, it could be supposed, that this fact can be explained by the absent of the certain enzymes’ competition for their substrate, but on the other hand the result may depend on the method of substances extraction.

In flax seeds the main part of sugars complex accounts for the so-called dietary fibers–the mucilage of the seed coat, endosperm starch and some other polysaccharides [31]. These fibers compose about 28% of the seed weight [17]. Mucilage, forming about 10% of the seed weight [38] consists of rhamnogalacturonan 1 (**RG1**), homogalacturonan (**HG**), arabinoxilan (**AX**), galactose and fucose residues, which may belong to **RG1** or **AX**, and a small amount of protein [26].

Using metabolomic profiling, we were unable to detect rhamnose, fucose and galacturonic acid (see below), that is, all pectins (**RG1** + **HG**). According to our previous data, during the first two hours of aqueous extraction, flax seeds secrete mucilage polysaccharides in the ratio: 51% of pectin (45**RG1** + 6**HG**), 4% of fucose, 26% of **AX**, 15% of **Gal** [39], which is about 2% of the seed weight. The loss of a whole group of substances could be a big disadvantage of the method, however, according to the latest data from other authors [40], these compounds belong to the densely packed outer layer of the cell wall of mucilage cells and in the absence of water could simply not be extracted from the seed coat. In addition, it can be assumed that **RG1** was precipitated by methyl alcohol and as a large molecule left during centrifugation. There is also another indirect evidence that fucose is a part of **RG**, and not **AX**. According to Miart et al. [40] the middle layer of the cell wall is represented by **AX**. Our new data show that the ratio **Ara**/**Xyl** = 3.14, and significantly exceeds the 0.23, described earlier [39]. This may indicate the presence of other polymers containing it, including arabinogalactan proteins [41]. The inner layer of mucilage cells is formed by xyloglucans and cellulose [40]. In flax, seed xyloglucans are mostly represented by XXXG repeats, and to a lesser extent XXGG [41], where X = **Glc**-**Xyl**, G = **Glc**, according to Fry et al. [42]. Perhaps, some part of the **Xyl** and **Glc** is accounted for xyloglucan. Contrary to our results, Salem et al. [43] in flax metabolom analysis were able to identify a high content of **Rha**, **Fuc** as well as **Xyl**, however, **Ara** was not detected. The leaders in content, as in our experiment, were **Raf** and **Suc**, but in comparison with them, the level of **Mylib** was two orders of magnitude lower. Perhaps, this is due to a more complex sample preparation of the material.

The endosperm is an intermediate tissue between the coat and cotyledons of flax seed, and most likely the main destruction occurs particularly in it during seed grinding and the release of starch, during the degradation of which **Malt** and **Glc** are formed. **Suc** and **Raf** are the main transport forms of sugars in the plant [44], **Glc**, **Fruct**, **Gal** and **Melib** may be products of the **Raf** degradation.

According to our data, flax seeds contain a lot of **Man** and about the same amount of **Ara**. Presumably, this can be explained by the presence of galactomannans, storage polysaccharides common for legumes [45]. Using size exclusion chromatography, it was shown that only trace amounts of **Gal** are present in the fraction of water-soluble polysaccharides of flax flour containing **Man** and **Glc** predominates [41]. After additional elimination of arabinogalactane proteins (**AGP**) from the same solution, the presence of rhamnose (2 mol%), **Ara** (28 mol%), **Xyl** (16 mol%), **Man** (9 mol%), **Gal** (29 mol%), and **Glc** (6 mol%) was revealed [41]. More than in a half of AGP, 6 **Man** residues are included in the structure of glycosylphosphatidylinositol lipid anchor [46], which may explain the detection of some **Man**. In addition, our results may be related to the presence of galacto-, glucomannans and arabinogalactan proteins in flour. However, these statements require additional confirmation.

### 2.3. Fatty Acids, Acylglycerols and Paraffinscomposition of Flax Seeds

Among the free fatty acids, acylglycerols and alkanes (paraffins), the largest amount (in total >76%) falls on α-linolenic (**lin**), linoleic (**lio**) and oleic (**ole**) acids (349, 339, 295 n.u., respectively), the main ones for linseed oil. From 2 to 7% is accounted for other free fatty acids–palmitic (**pal**), stearic (**ste**), gamma-linolenic (**glin**), vaccenic (**vac**) acids; monoacylglycerol 2 (**MAG2**) and diacylglycerol (**DAG**), which include **lio** (94, 51, 41, 25, 35, 33 n.u.). Less than 1% is accounted for minor components of this group of acids–pelargonic (**pel**), lauric (**lau**), eicosanoic (**eic**), lignoceric (**lign**), monoacylglycerols 1 and 3 (**MAG1** and **MAG3**), as well as paraffin C18 (**parC18**) −1.0, 0.29, 7.8, 7.1, 8.4, 6.5, 1.7 n.u., respectively (Figure 3c,d, Table 1 and Appendix A).

Significantly more free fatty acids, acylglycerols and **parC18**, in comparison with the other lines, have the yellow-seed lines gc-136 (**pal, lio, ole, ste, glin, MAG1, 3**), gc-391 (**ole, eic, MAG2, DAG**), gc-132 (**ole, ste, MAG2**), gc-159 (**pel, vac, parC18**), gc-173 (l**au, MAG1, parC18**), gc-141 (**lign**); significantly less amount of these compounds has the line gc-109 with brown seeds and white flowers. Some of the lines have a minimum amount of one or more of the detected substances listed above: gc-2 (**pel, lau, lign, MAG3, DAG, parC18**), gc-65 (**vac, glin, MAG1, 2, DAG, parC18**), gc-121 **(vac, MAG2, 3**), gc-391 (**lin, glin**), gc-159 (**lau**), gc-141 (**lin**), gc-210 and 473 (**vac**), gc-432 (**lign**), gc-129 (**eic**), gc-124 (**MAG3**), gc-132 (**parC18**) (Appendix A).

Correlation analysis of fatty acids content in flax seeds showed the presence of a large pleiad, including closely related **pal, ste, ole, lio**, **α** and **γ lin**, **MAG1** and **3**. The second pleiad is formed by **pel** and **vac** acids, as well as **parC18**. The third pleiad includes **eic**, **MAG l, 2** and **DAG**. The second pleiade is related to the first one due to correlations of **pel** and **vac** with **pal**, and also by correlation between **vac** and **ste** acids. The third pleiade is related to the first pleiad due to the correlations of **eic** and **MAG2** with **lio**, **lau** and **lign** were found to be independent of the others. As in the case of sugars, all reliable correlations are positive (Figure 2c, Appendix A).

The ratio of main fatty acids in flour and in oil differs significantly. From linseed flour compared with oil some acids are extracted in a larger ratio: **pal** (9 and 5%, respectively), **ste** (4.5 and 3.9%), **ole** (30 and 24%), **lio** (28 and 21%), and in much lower ratio–**lin** (29 and 47%) (Appendix A). These discrepancies can be explained by different methods of extraction of target substances used in the analysis of metabolomic profiles and fatty acid composition of oils [47].

Differences in the relative content of fatty acids extracted from flour and oil were found by Tavarini et al. [48] when evaluating the Bethune cultivar, characterized by a high content of **lin** in oil. It turned out that the flour from the seeds of this cultivar contains even more **lin** then oil had, and less **ole** and **lio**. However, in the low-linolenic cultivar Solal, these ratios are similar.

In absolute values, the content of both major and minor fatty acids extracted directly from the flour, according to our results, is 10 times less than it was described by Tavarini et al. [48] or 200 times less than it was isolated from oil when converted to flour weight by Sargi et al. [49]. It happened because during the analyses of metabolomic profiles, only free fatty acids are extracted.

### 2.4. Polyatomic Alcohols Composition of Flax Seeds

Among polyatomic alcohols, the largest amount falls on **glycerol**, inositol (**Inosl**) and galactinol (**galtl**) forming in total more than 76% of sample (141, 132, 67 n.u., respectively). From 1 to 6% is accounted for other alcohols-myo-inositol (**mInosl**), erythritol (**erythl**), xylitol (**xyltl**), glycerol-3 phosphate (**glyclphs**), mannitol (**mannitol**), dulcitol (**dulcl**) (27, 27, 19, 16, 12, 5 n.u., respectively) (Figure 3e, Table 1 and Appendix A).

Significantly more polyatomic alcohols in comparison with other genotypes contain gc-136 (**glycerol, glyclphs, mannitol**), as well as gc-159 (**erythl, glyclphs**), gc-432 (**xyltl, Inosl**), gc-391 (**dulcl**), gc-132 (**mInosl**), gc-173 (**mInosl**), gc-473 (**mInosl**) and gc-65 (**galtl**). Significantly less amount of alcohols in comparison with the others is present in gc-2 (**erythl, glycerol, mannitol**) and in gc-109 (**glycerol, glyclphs, xyltl, dulcl, mInosl, Inosl**), as well as in gc-391 (**erythl, mannitol, galtl**), gc-121(**glycerol, dulcl**), gc-65 (**glyclphs, dulcl**), gc-210 (**glyclphs, mannitol**), gc-159 (**xyltl, mannitol**), gc-124 (**mannitol**), gc-141 (**dulcl**), gc-473 (**galtl**) (Appendix A).

Correlation analysis showed the presence of one friable pleiad, where **mannitol** is closely related to **glycerol**, which may be due to the peculiarities of these compounds accumulation in flax seeds, which requires additional evaluation. **Mannitol** is moderately correlated with **xyltl**, which, in turn, is associated with **Inosl**. **Mannitol** is associated with **glyclphs**, which strongly correlates with **erythl** and moderately with **mInosl**. **Dulcl** and **galtl** are independent of other alcohols (Figure 2d, Appendix A).

Salem et al. [43] in flax metabolomic analysis showed approximately the same ratio of basic polyatomic alcohols as it occurred in our experiment, when glycerol, **galtl**, **mInosl** (and possibly **Inosl** similar to it) showed order of magnitude greater amounts than **erythl** and **xylt**. They did not identify mannitol and its stereoisomer **dultl**, but isolated their stereoisomer sorbitol.

There is practically no other data about content of polyatomic alcohols in flax seeds. The high content of glycerol and glycerol-3 phosphates can be, among other reasons, explained by the decay of lipids into glycerol and free fatty acids. **Inosl** is a part of the glycolipids of the membrane, being an anchor for polysaccharides and the carbohydrate part of glycoproteins; it participates in signaling [45] and possibly in the formation of oleosomes. **Galtl** consists of **mInosl** and a galactose residue and serves as a donor of the latter substance during the biosynthesis of **Raf** from **Suc** [44], the main oligosaccharides isolated in our analysis. Flax seeds are rich in phytic (myo-inositol hexaphosphate), which accumulates about 70% of all phosphorus in defatted flax flour [50]. This acid was not detected in our experiment, but we can suppose that it was fragmented during sample preparation into **mInosl** and phosphoric acid.

**Mannitol** and, possibly, its stereoisomer **dulcl**, belong to osmolytes, substances that increase the viscosity of the cytoplasm and absorb reactive oxygen species, thereby protecting the cell from free radicals, maintaining the integrity of membranes and metabolic activity of tissues, which ensures the resumption of growth after improving the water regime after drought [51]. **Erythl** is one of the precursors for the formation of aromatic substances in the shikimate pathway [44].

### 2.5. Secondary Metabolites Composition of Flax Seeds

The combined group of “secondary metabolites” contains phenolic substances: vanillic acid (**vanilic**), **coniferol**, **caffeic** and **ferulic** acids, kaempferol (**kaemph**); heterocyclic aromatic substances: quinoline (**quinolin**) and isoquinoline (**Iquinolin**); phytosterols: sitosterol (**sitostrl**) and campesterol (**campstrl**), belonged to biologically active compounds. Some of the secondary metabolites are attributed to other groups of substances (for example, salicylic acid belongs to organic acids, etc.) and are not described here.

In the group under consideration, the largest amount (in total >64%) falls on **ferulic** and **sitostrl** (96, 63 n.u., respectively). From 6 to 15% falls on **campstrl**, **kaemph** and **coniferol** (38, 26, 16 n.u., respectively) and less than 4% accounts for **quinoline**, **Iquinolin**, **caffeic** and **vanillic** (3.6, 2.5, 0.89, 0.46 n.u.) (Figure 3f,g, Table 1 and Appendix A).

Significantly more secondary metabolites in comparison with other genotypes have gc-432 (**sitostrl, caffeic, quinoline**), gc-136 (**ferulic**) and gc-473 (**caffeic**), as well as gc-391 (**campstrl**), gc-119 (**coniferol**), gc-2 (**kaemph**), gc-132 (**vanillic, ferulic**), gc-159 (**Iquinoline**). Significantly less than the others secondary metabolites are present in gc-109 (**campstrl, coniferol, Iquinoline**), as well as in gc-391 (**sitostrl, kaemph, vanillic, quinoline**), gc-159 (**coniferol, caffeic**), gc-2 (**vanillic, Iquinoline**), gc-173 (**coniferol**), gc-119 (**kaemph**), gc-132 (**kaemph**), gc-65 (**ferulic**), gc-210 (**vanillic**) (Appendix A).

Correlation analysis showed the presence of two independent pleiades. The first one is formed by positively correlating **quinoline**, **sitostrl** and **caffeic**, the latter one is associated with **coniferol**, and **quinoline** negatively correlates with **Iquinoline**. The second pleiad is formed by **vanillic** and **ferulic**. **kaemph** and **campstrl** are independent of other members of this group (Figure 2e, Appendix A).

The extraction method we used turned out to be more effective/sensitive for the determination of one of the flavonoids–**kaemph**, than the spectrophotometric one, since its content in our experiment was on average 5 times higher than that reported by other authors (26 and 5.4 n.u., respectively) [51]. According to the bibliography data, in flax seeds flavonoids herbacetin glycosides are also present, but they were not detected by the method we used [32,33].

According to our data, the amount of **ferulic** was about 9 times higher than previously noted (96.2 and 10.9 n.u., respectively) [Harris, Haggerty 1993, cited by 51]. However, according to the total content of phenols, the spectrophotometric method (Price, Butler, 1977, cited by [52]) revealed two orders of magnitude higher values than that detected in our experiment. Possibly it happened due to phenols association with other substances, or undetectable lignans, as well as phenylpropanoid glycosides (linusitamarin, linocinamarin), which was confirmed by the works of Wang et al., [53], who spectrophotometrically detected both free and bounded phenols. It is shown that **caffeic** is detected in approximately equal proportions in both forms, para-coumaric acid has 4 times more bounded forms than free ones, and **ferulic** has this difference of 3–15 times.

Probably, only a small amount of phenolic compounds passes from seeds to oil during cold pressing. Herchi et al. [34] showed that the main phenolic compound in oil is **vanilic**, there are also **PHBA** (see organic acids below), **ferulic**, its methyl ester, methyl ester of **cinnamic** acid, as well as **lignans** (0.155, 0.029, 0.022, 0.014, 0.012, 0.013, 0.11 n.u., respectively). So, **vanilic** is the best to transfer into oil, where its amount is only 3 times less than in the whole seeds. In addition, in comparison with intact seeds, oil contains 2 orders of magnitude less of **PHBA** and 4 orders of magnitude less **ferulic**.

Interestingly, using classical biochemical analysis, it was shown that flax seeds contain the dimer of coniferyl alcohol–secoisolariciresinol in a large amount [33], which belongs to the class of lignans, precursors of phytoestrogens that cause the medicinal properties of flax seeds [18,21]. It can be assumed that the **coniferol** identified in our experiment also was partially obtained after the destruction of lignans during samples preparation.

### 2.6. Amino Acids and Other Nitrogen-Containing Compounds Composition of Flax Seeds

In our experiment, 12 out of 20 proteinogenic amino acids were identified. Among the free amino acids and other nitrogen-containing compounds (**carbamide**), the largest amount (in total >56%) falls on aspartic acid (**Asp**), proline (**Pro**), glutamic acid (**Glu**), alanine (**Ala**), (21.0, 20.0, 19.6, 14.2 n.u., respectively). From 5 to 9%–falls on hydroxyproline (**Hyp**), glycine (**Gly**), phenylalanine (**Phe**) and valine (**Val**)–12.5, 10.4, 9.8, 6.9 n.u., respectively, and less than 4%–for threonine (**Thr**), leucine (**Leu**), serine (**Ser**), tyrosine (**Tyr**), histidine (**His**), ornithine (**Orn**), gamma-aminobutyric acid (**GABA**), and carbamide (**carbam**): 6.0, 4.7, 2.7, 2.5, 1.1, 0.9, 0.5, 0.6 n.u., respectively (Figure 3h, Table 1 and Appendix A).

Significantly more free amino acids in comparison with other genotypes have the black-seeded line gc-432 (**Ala, Gly, Hyp, Val, Leu, Thr, carbam**) and yellow-seeded gc-173 (**Pro, Glu, GABA**), as well as gc-141 (**Ser, Thr, Asp, Glu, Tyr**), gc-119 (**Orn, Phe, His**), gc-136 (**Pro**), gc-121 (**Asp**), gc-473 (**His**). Significantly less amount of other amino acids is present in gc-109 (**Val, Ser, Orn, Glu**) and gc-391(**Pro, Hyp, Val, Ser, Thr, Asp, carbam**), as well as in gc-159 (**Ala, Orn, Tyr**), gc-210 (**Gly, GABA, His**), gc-121 (**Leu, Orn**), gc-129 (**Val, Ser, Thr**), gc-2 (**GABA, Phe**), gc-119 and 124 (**Glu**), gc-132 and 136 (**carbam**), gc-432 (**Orn**) (Appendix A).

Correlation analysis revealed a large pleiad, including the closely positively correlated **Hyp**, **Val**, **Leu, Gly, Ala** and **carbam**. The second pleiad is formed by **Ser**, closely related to **Thr** and **Tyr**, the latter correlates with **Thr**, and also with **Val**, **Leu**, **Gly** and **carbam** from the first pleiad. **Ser** is associated with **Val** from the large pleiad, as well as with **Glu**. **Pro** is bound on the one hand to **Ser** and **Tyr**, and on the other hand to **GABA**, which in turn is bound to **Glu**. **His** is closely correlated with **Orn**, which in turn is related to **Tyr**. **Phe** correlates only with **Val**. **Asp** is independent of other amino acids (Figure 2f, Appendix A).

Salem et al. [43] in flax metabolom analysis identified 19 out of 20 proteinogenic amino acids. The leaders, such as in our experiment, were **Pro**, **Ala**, **Asp**, as well as tryptophan, which we did not identify.

Only free amino acids are extracted by methanol, so their absolute amount is less than that obtained with the use of the standard technique, for the amino acid composition of proteins detection (2.5% compared to 10–33%, respectively). From the qualitative point of view, the ratio of aspartic acid (**Asp**), threonine (**Thr**), valine (**Val**), phenylalanine (**Phe**) was the same; in 2–3 times less amounts were detected for glutamic acid (**Glu**), leucine (**Leu**), serine (**Ser**), histidine (**His**) and in 2–3 times more of proline (**Pro**) and alanine (**Ala**). We did not identify cysteine, lysine and arginine, and only a few lines showed the presence of methionine and isoleucine (Appendix A) [24,29].

The presence of a large amount of free oxyproline may serve as indirect evidence of the presence in flax seeds of structural proteins with a high level of glycosidation, including arabinogalactans and proline-rich proteins [54].

Flaxseed is the sources of cyclic peptides called cyclolinopeptides, for which about 25 different kinds are known. For example, cyclolinopeptide A (cyclo-(Pro-Pro-Phe-Phe-Leu-Ile-Ile-Leu-Val) has been shown to have antimalarial, immunosuppressive and other activities (Bell et al., 2000 cited by [55,56]). However, cyclolinopeptides may be undesirable in flax edible oil, as they give it bitterness and in technical, as they reduce the drying rate of oil [56]. Perhaps part of **Pro**, **Phe**, **Leu** and **Val**, originate from these peptides.

Another group of substances typical for flax–cyanogenic glycosides–was not identified. These are anti-nutritional substances consisting of **Val** and isoleucine residues, a nitrile group, as well as one or two **Glc** residues [31]. It can be supposed that this method of analysis does not determine them.

### 2.7. Organic Acids Composition of Flax Seeds

In the group of organic acids (organic and phosphoric acids), the largest amount (in total >73%) falls on phosphoric, glucono-1,5-lactone (**gluALactn**), malic (**malic**), azelaic (**azelaic**) acids (40.1, 10.8, 10.2, 8.7 n.u. respectively). From 4 to 1% is accounted for tartaric (**Tartaric**), oxalic (**oxalic**), lactic (**lactic**), ribonic (**ribonic**), gallic (**gallic**), succinic (**succinic**), gluconic (**gluconic**), 4 hydroxybenzoic (**PHBA**), L-iduronic (**LdPUA**), methyl phosphate (**methph**) and 3 hydroxypropionic (**3hydroxypr**) acids and less than 1% accounts for **threonic**, methylmalonic (**methylmalonic**), **pyruvic**, **glyceric**, **Salicylic**, 5-hydroxypipecolic (**5hydroxypipecolic**), **benzoic** and **nicotinic** acids (Figure 3i,j, Table 1 and Appendix A).

Four of the acids described above also belong to phenolic compounds (**benzoic, PHBA, Salicylic, gallic**), but traditionally they are analyzed together with the other acids, in contrast with oxycoric acids (**caffeic** and **ferulic**), which are described together with phenolic compounds.

Significantly more organic acids, in comparison with other genotypes, have gc-432 (3hydroxypr, PHBA, ribonic, gluALactn, gluconic), as well as gc-159 (lactic, pyruvic, phosph, benzoic, nicotinic, succinic, glyceric, Salicylic, Threonate), gc-173 (methph, succinic, Tartaric, ribonic, gluconic), gc-119 (oxalic, malic), gc-210 (5hydroxypipecolic, gallic), gc-2 (methylmalonic), gc-473 (methylmalonic, azelaic), gc-124 (methph), gc-141 (PHBA) and gc-65 (IdPUA). Significantly less than in seeds of other lines organic acids were detected in gc-109 (lactic, methylmalonic, oxalic, methph, nicotinic, succinic, glyceric, 5hydroxypipecolic, malic, Salicylic, Threonate, ribonic, gallic) and gc-391 (3hydroxypr, phosph, 5hydroxypipecolic, Threonate, Tartaric, azelaic, gluALactn, IdPUA), as well as gc-65 (pyruvic, oxalic, PHBA, nicotinic, succinic, glyceric, 5hydroxypipecolic, Salicylic, gluconic), gc-2 (pyruvic, oxalic, benzoic, PHBA, nicotinic, 5hydroxypipecolic, Threonate), gc-132 (gallic, ribonic),gc-121(pyruvic) and gc-129 (azelaic) (Appendix A).

Correlation analysis showed the presence of a large main pleiad, which included closely positively correlated **ribonic**, **phosph**, **Salicylic, Threonate, lactic, pyruvic, glyceric, succinic** and **benzoic** acids, with most of these acids positively correlated **Tartaric** and **azelaic** acids. This big pleiad is connected with other one, including **oxalic** and 3 hydroxypropionic acid (**3hydroxypr**) due to the correlation with **ribonic**, as well as some other acids. 4 hydroxybenzoic (**PHBA**), **gluconic** and **gluALactn** form a pleiad, which is associated with **azelaic** and **ribonic**, as well as some other acids from the first pleiad. **malic** acid connects these two pleiades due to correlations with all members of the second pleiad, as well as with **gluALactn**. Pair of closely correlated **gallic** and 5 hydroxypipecolic (**5hydroxypipecolic**) acids, as well as methylphosphoric acid are relatively independent from the others. Completely independent of others are **methylmalonic** and L-idopyranuronic acids (Figure 2g, Appendix A).

Our method proved to be more productive in the isolation of organic acids and allowed the identification of 23 substances, compared to that of Salem et al. [43] which identified16 substances of which 6 coincided (**phosph**, **malic**, **oxalic**, **succinic**, **purivic**, **threonate**). The ratio of all of them except **threonate** appeared to beproportional to our data, which was much larger.

Organic acids are a very heterogeneous group of substances. Inorganic phosphoric acid–is the residues of phytic acid’ destruction [35], and organic one–is methyl phosphoric acid–possibly formed after the destruction of phospholipids. **Azelaic** is a derivative from **ole**, it is a mobile molecule involved in the formation of long-term resistance (systemic disease resistance) [57]. **Malic, succinic, methylmalonic** and **pyruvic** acids are participating in the Krebs cycle. **Lactic** acid is the result of glucose conversion by anaerobic glycolysis or fermentation [44]. **Tartaric**, **oxalic** and **Threonat** can be a products of ascorbic acid oxidation, and the first two of them in the light can also be its precursors [58,59]. Some organic acids belong to phenolic compounds (secondary metabolites): **benzoic**, **PHBA**, **Salicylic** and **gallic**. The content of the latter one in our experiment practically agreed with the previously obtained results (Harris and Haggerty 1993, cited by [51]). These acids have protective functions: **benzoic** and **PHBA**–as preserving agents, **gallic**–as part of hydrolysable tannins, **Salicylic**–acts as a phytohormon of a nonspecific reaction to stress (damage of the integumentary tissues of the plant, etc.). Cyclic nitrogen-containing compounds include **5 hydroxypipecolic** and **nicotinic** acids. The first one is a mobile metabolite which induces a long-term resistance (systemic disease resistance) [59], the second one is a part of the coenzymes NAD and NADP of the majority of reduction–oxidation reactions [44].

The accumulation of **gluconic** during the oxidation of **Glc** in plant tissues occurs through an intermediate compound **gluALactn**, and when the primary alcohol group of glucopyranose is oxidized, glucuronic acid is formed [60]. According to Singh and Kumar [61], gluconic plays an essential role in protecting plant tissues from heavy metal ions through the formation of chelates. Glucopyranuronic acid is a part of pectins (structural and protective, in particular from abiotic stresses) plant cell compounds [62].

### 2.8. Factor Analysis of Flax Seeds Chamical Composition

Factor analysis revealed 6 factors describing more than 70% of the total detected compounds variability (Figure 4, Appendix A).

The first one is the sugar content factor, mainly it affects sugars (**Xyl, OgliDgal, Sorb, Fruct, Ara, Tur, Rib, Gal, Suc, Raf, Glc**) and related sugar alcohols (**Inosl, xyltl**), and organic acids (**gluALactn, gluconic**), as well as other organic acids (**hydroxypr, malic**). This factor also has a great influence on **carbam**, free amino acids–proteinogenic aliphatic (**Leu, Ala, Val, Gly**), oxycarboxylic (**Thr**), as well as hydroxyproline (**Hyp**), quinoline, phenolic (**caffeic, PHBA**) substances and **sitostrl**.

According to this factor, the undoubted leader is the gc-432–black seed line with the maximum values of the characters affected by this factor.

The second factor can be defined as the acid factor. It affects most of organic (**benzoic, glyceric, pyruvic, Threonate, nicotinic, lactic, Salicylic, phosph, succinic, ribonic, Tartaric, azelaic**) acids, as well as some free fatty non-“storage” (**pel, vac, lign**) acids, **glAld, parC18, Asp, erithl, glyclphs** and **Iquinoline**.

According to this factor, one line is also the undoubted leader–it is the yellow-seeded gc-159 with maximum values for each of the characters affected by this factor.

The third factor is the factor of lipid metabolism compounds (“storage” substances) and stress resistance, it affects free fatty acids and substances related with them (**lin, glin, pal, ste, lio, ole, MAG1, MAG3, glycerol, glyclphs**), **mInosl**; substances responsible for stress resistance (**Pro, GABA, mannitol**).

This factor recognized three lines–gc-132, 136 and 173. All of them had the maximum content of **Pro**, **ferulic** and substances involved in lipid metabolism. Gc-136 and 173 had the maximum content of **glyclphs** and **GABA**, gc-132 and 173-**mInosl**, gc132–**vanilic**, and gc136–**mannitol**.

The fourth factor is the factor of polar free amino acids, which affects the content of these amino acids (**Ser, Tyr, Asp, Pro, Glu, Thr**) and **GABA**, some organic (**methph, tartaric, gluconic, gallic, azelaic, 5hydroxypipecolic**) and free fatty (**Lau, Lign**) acids, as well as sugars (**Melib, Man, Glc**). This factor identified three lines–gc-141, 173 and 473. All of them had the maximum content of **Ser, Tyr, Thr, methph**, and **azelaic**. Gc-141 and 173 had the maximum content of **Glu**, Tartaric and **gluconic**, as well as **Melib**, gc-141-**Asp** and **lign** acids and **Glc**, 173–**GABA** and **lau**, as well as **Man**.

The fifth factor-the factor of phenolic compounds affects their content (**coniferol, vanillic, kaemph, Phe**), as well as **Hyp**, **His**, **Orn**, **Lyx**, dicarboxylic acids (**oxalic, malic**), glucose-containing sugars (**methGluD**, **Malt, Tur**), or inositol alcohols (**mInosl, galtl**). Interestingly, according to this factor, two substances–**kaemph** and **galtl** showed antagonism to the rest of the characters, possibly because **kaemph** is a competitor of conifer and vanilla alcohols in the phenylpropanoid biosynthesis pathway, and **mInosl** is a decay product of **galtl**. This factor showed cardinal differences of gc-119 and 473 from gc-65. Gc-119 and 473 had the highest, and gc-65–the lowest content of **His**, **Orn**, **Phe, Lyx, methGluD, Malt** and **Tur**, and vice versa gc-119 and 473 had the lowest, and gc-65-the highest content of **galtl**. Gc-119 had the highest, and gc-65–the lowest content of **coniferol**, **Hyp** and **oxalic**. Gc-119 and 473 had the highest content of **mInosl**, and gc-119-**malic** and the least **kaemph**.

The sixth factor is the factor of large “not completely destroyed” substances. It determines the content of acylglycerols (**DAG, MAG2**), **eic**, **campstrl**, as well as **dulcl** and **gallic**. Some of the substances: **IdPUA**, **galtl** and to a lesser extent **Suc** and **Raf** occupy the opposite position in terms of loads for this factor.

This factor showed the cardinal differences between gc-391 and gc-65 (the smallest and the largest loads, respectively). Gc-391 had the highest, and gc-65–the lowest content of **dulcinol**, **DAG**, **MAG2**, **eic**, and vice versa, gc-391 had the lowest, and gc-65–the highest content of **IdPUA** and **galtl**. Gc-391 had the highest **campstrl** content, while gc-65 had the lowest **gallic** content.

Thus, factor analysis made it possible to differentiate all 16 lines, of which 10 occupied a separate position by one or two factors. Interestingly, the first two factors with the highest loads (20 and 15% of the total variability) showed an isolated position of two lines. The seeds of the first–gc-432 are even outwardly very different from that of the other lines, while the second one–gc-159 has gained its position, differing from the rest in the complex of organic acids and other substances that make up about 1% of the identified constituent in the seed in total.

### 2.9. Association of Plants Morphological Features with the Chemical Composition of Flax Seeds

Despite the fact that the study included lines covering the maximum intraspecific diversity in terms of morphological characteristics of seeds, it turned out to be possible to form several more or less homogeneous groups based on flower and seed coloration in a sampling of 16 lines: having the color of “wild type” seeds–red-brown (6 lines), yellow seeds (6) and white corolla (4). Using the Student’s t-test and the Mann-Whitney U-test, it was found that lines with brown seeds have on average less **Glu**, lines with yellow seeds have more **Iquinoline**, and white-flowered lines have less **3hydroxypr**, **phosphoric**, **ribonic**, **Asp** and total sum of organic acids, **sitostrl** and **Raf** (Table 2). These data can’t be interpreted as a rule, but it can serve as a basis of future investigations.

Despite the apparent differences between yellow and brown seeded genotypes, we found only one publication analyzing the complex divergence of substances in them. Salem et al. [43] evaluated the differences between 4 brown-seeded and one yellow-seeded varieties of linseed grown in three locations in Egypt. The authors showed that the yellow-seeded variety Sakha 6 accumulated significantly higher amounts of several amino acids, (**His**, **Ala**, **Gly**, Arg, **Ser**, Met, **Val**, **Tyr**, **Phe**, **Gln**), but the total protein content did not differ significantly among the tested varieties. Sakha 6 seeds showed high level of free organic acids (citric, glutaric, **pyruvic**, and fumaric) and fatty acids (**lio**, **lin**). Several sugars (**Glc**, **fruct**, rhamnose, **Xyl**, **Raf**) showed low levels in the Sakha 6 cultivar compared to brown seeded cultivars. Secondary metabolites (pantothenic, sinapic, cinnamic and benzoic acids, riboflavin, glutathione) have been accumulated in Sakha 6 seeds in significantly higher concentrations, then in the rest genotypes. Linustatin and neolinustatin showed low levels in the Sakha 6 cultivar compared to other genotypes.

In our experiment we did not identify such regularities in the entire group of the tested yellow-seeded lines. However, some of the sampling members were distinguished by these substances: yellow-seeded lines gc-173 (max **Ala**, **Val**, **Tyr**, min **Xyl**), gc-159 (max **pyruvic**, **benzoic**, min **Xyl**), gc-136 (max.**lio** and **lin**, min **Glc**, **Fruct**, **Xyl**), gc-391 (min **Glc**, **Fruct**). Thus, even among the small group of yellow-seeded lines of the VIR collection, there is a great variability of biochemical characteristics.

Interesting guesses can be taken from the work of Ramsay et al. [32], who studied the metabolomic profiles of three flax groups differing in the level of linolenic acid. The biosynthesis of linolenic acid has been well studied and is carried out by desaturases encoded by two genes *LuFAD3A* and *LuFAD3B* [63], which have no other activities. It is known that low-linolenic flax varieties are used for food purposes and have yellow seeds. Ramsay et al. [32] identified that low-linolenic forms contain significantly less flavonoid herbacetin diglucoside and more glucoside of coumaric, caffeic and ferrulic acids. Interestingly, in an earlier publication, Struijs et al. [64] it was shown that these substances can be a part of one lignan macromolecule from flaxseed hulls. It is most likely that the inhibition of anthocyanin biosynthesis also affected the herbacetin diglucoside flavonol, which was part of the lignan macromolecule through the formation of glycoside bonds [64]. It is possible that glycosides of oxycoric acids are added in its place, which leads to an increase in their concentration.

## 3. Materials and Methods

### 3.1. Material

In this case, 16 lines of the sixth generation of inbreeding, created in VIR department of genetic resources of oil and fiber crops were used in the experiment (Table 3). The gc-2 line acted as a “wild type”. Most of other lines (gc-65, 432, 121, 124, 129, 136, 141, 159, 173, 391) had different seeds color, the genetic control of which was described earlier [65]. Gc-391 was selected from a low-linolenic variety used for food. Gc-119 has medium content of linolenic acid. It was created on the basis of an Indian accession originating from the region where the production of flax flour is widespread. Mutations of some lines could theoretically affect the characters of the seed embryo (complete inhibition of the anthocyanin coloration of the entire plant in gc-136; pigmentation disruption in all organs except the seed in gc-132, 391, reduction of the chlorophyll coloration of the plant in gc-210, 473).

### 3.2. Chemicals and Reagents

Methanol and chloroform are of high-performance liquid chromatographic grade obtained from Merck KGaA (Darmstadt, Germany), pyridine, tricosane, N,O-bis(trimethylsilyl)trifluoroacetamide obtained from Sigma-Aldrich (St. Louis, MO, USA), Standard Certified Reference Material: alkane standard C7-C40 in hexane,1 mL obtained from Millipore Sigma Supelco (49452U, Saint Louis, MO, USA).

### 3.3. Samples Preparation and Metabolomic Analysis

Each sample was represented by 10 g of seeds mix. Samples preparation followed the protocol published by Perchuk et al. [12,66] with minor modifications. Seeds of the lines approximately 50 mg, homogenized with cold methanol in a ratio of 1:10 (*w*/*v*) on Vortex Genius 3 (IKA, Staufen im Breisgau, Germany). Samples were infused for 30 days at 5–6 °C [67], and then resulting extract centrifuged (4 °C, 14,000 rpm for 10 min) on centrifuge 5417 R (Eppendorf AG, Hamburg, Germany). 100 µL of the supernatant was evaporated on a CentriVap Concentrator (Labconco, Kansas, MI, USA); then silylated with 50 µL bis(trimethylsilyl)trifluoroacetamide for 30 minutes at 100 °C on Digi-blok (Laboratory Devices, Inc., Hollison, MA, USA). The separation of substances was carried out on an HP-5MS capillary column (stationary phase: 5% phenyl, 95% methylpolysiloxane, 30.0 m, 250.00 microns, 0.25 microns, Agilent Technologies, Palo Alto, CA, USA) using Agilent 6850 gas chromatograph with an Agilent 5975B VL MSD quadrupole mass-selective detector and auto-injector G2613 A (Agilent Technologies, Santa Clara, CA, USA) controlled by Agilent Chem Station E.02.02.1431 software (Agilent Technologies Inc.). Conditions of the analysis: the speed of helium through the column is 1.5 mL/min (pressure: 0.834 bar), heating from 70 °C to 320 °C, at a speed of 4 °C per minute. The injector temperature is 300 °C at splitless mode, the injection volume is 1 µL two washing cycles with chloroform solvent carried out after each injection process (60 µL each). The internal standard is a solution of tricosan (RI = 2288.0) in pyridine (1 µg/µL). The peaks were recorded by quadrupole mass-selective detector Agilent 5975B VL MSD. Electron impact ionization was performed at 70 V and ion source temperature of 250 °C. The quadrupole was operated in a scanning mode in the m/z range from 50 to 550 with scanning at 0.34 sec scan-1.

Substances were identified by their mass spectrum and retention index (RI) using the AMDIS program (Automated Mass Spectral Deconvolution and Identification System, National Institute of Standards and Technology, USA, http://www.amdis.net, accessed on 2 October 2020). For the results analyses were used: libraries NIST 2010 (National Institute of Standards and Technology, USA, http://www.nist.gov, accessed on 2 October 2020), and in-house libraries of the Resource Center “The Development of Molecular and Cell Technologies” of the St. Petersburg University and Botanical Research Institute RAS (St. Petersburg, Russia) [10,66]. These last two databases were developed as the result of previous standard-based chemical determination performed at St. Petersburg University and the Botanical Institute of the Russian Academy of Sciences The substance was considered as identified when its index of substance’s mass spectrum matching with a library variant with at least 80.

The retention indices (RI) were estimated by the calibration of saturated hydrocarbons with the number of C atoms ranging from 10 to 40. A semi-quantitative assay of the metabolite profiles was performed by calculation of the total ion peak areas with the internal standard method using UniChrom software (UniChrom TM 5.0.19.1134, New Analytic Systems LLC, Belarus, www.unichrom.com, accessed on 2 October 2020).

A total of 295 peaks were marked, of which about 90 substances were identified. The substances relative content is given in in nominal units (n.u.) (Table 1).

### 3.4. Statistical Analysis

Correlations between characters (Pearson correlation coefficients) were calculated according to the standard methods [68,69] using packet of programs MS Excel and Statistica 7.0 for Windows. Correlations 0.7 > r ≥ 0.5 were considered as intermediate ones, 0.9 > r ≥ 0.7 as high, and r ≥ 0.9 as very strong correlations. Correlation pleiades were composed manually by MS Word [70].

Principal component method of analysis was used to extract factorial load. Eigenvalues, percent of the total variance and cumulative percentage share of the extracted factors were calculated. Selection of factors number was carried out using the scree-test, when the smooth decrease of eigenvalues appears to level off to the right of the plot [71]. To acquire the simplest structure of factor loadings, factor rotation method (Varimax raw) was used. Rotation cause for each substance has maximum load only of one factor and for other factors loadings are approximated to zero [26,71]. To determine the significance of the factor influence on the character, the square root of factor variance fraction according to the formula: Significance of the factor influence = (% of factor variance/100)^0.5^. If more than one significant factor load for one character, the analysis excluded those which were 0.2 less than the maximum one [26]. For lines’ classification, factor scores were used. Among all possibilities of pair-factor systems, four pictures were chosen in order to (i) show all 6 factors and (ii) indicate the most evident grouping of characters and lines.

In the text abbreviated name of evaluated characters are marked in bold, see Table 1.

## 4. Conclusions

There is no universal method that isolates all substances simultaneously in a real ratio from biological objects. Definite methods extract compounds with different intensities depending on their availability and a number of other factors. Many substances are interconnected with each other and the extraction of one can slow down the output of the others, or vice versa accelerate it.

For this pilot study, the lines presenting the maximum intraspecific diversity in terms of morphological characteristics of seeds were selected. These lines showed a wide variability of 90 biochemical compounds. The analysis showed a significant difference between the data obtained in our experiment and the results of extractions, carried out by other methods. Sugars are relatively easier to isolate, and fatty acids are much worse. Protein molecules are difficult to break down and therefore very few free amino acids were released. No chemical extraction can be compared with mechanical extraction of fatty acids, especially it refers to linolenic acid. In spite of the good extraction of sugars, pectins, including rhamnogalacturonan 1, as well as fucose, are poorly isolated, but a lot of galactomannans are detected. There are only scattered data about polyatomic alcohols, organic acids and secondary metabolites. Lignans are extracted definitely worse, or rather their residues of them after sample preparation (coniferol). Phytin does not stand out at all (it may have disintegrated during sample preparation releasing myo-inositol and phosphoric acid) and cyanogenic glycosides, which are the “calling card” of flax seeds. The method detected many secondary metabolites related to phenols, phytosterols and heterocyclic aromatic compounds, but it is the most effective in determining one of the flavonoids–kaempferol. Organic acids are a very well-represented class of compounds from qualitative point of view, possibly due to the ease of their extraction. In general, in our opinion, the method is the best for isolation of free substances, as well as those directly contacting with membranes (proline, hydroxyproline, inositol, glycerol, phosphates, etc.).

As it was noted above, metabolomic profiling proved to be weakly effective in determining relatively complex biologically active substances (BAS) typical of flax-mucilage (especially pectin part), linolenic acid, phytin, lignans. However, it opened up the possibility of determining easily extractable BAS.

Factor analysis, developed just for such multifaceted studies, identified six factors, most of which coincided in influence with the groups of substances or compounds with similar chemical properties/metabolic pathways that we identified. That is, despite of the method disadvantages, each of the highlighted groups of substances is allocated in a proportional amount from flax seeds of different genotypes.

The method of metabolomic profiling is valuable because it allows to differentiate samples by a set of characters, which is confirmed by the data of factor analysis, which uniquely identified 10 out of 16 lines. The use of these results can help in the selection of the initial material for breeding aimed to formation of a complex of economically valuable traits. In addition, the method we have applied is ideally suitable for the primary screening of biological diversity by a set of characters and structuring of plant collections. To identify physiological and biochemical patterns, it is necessary to find a balance between the accuracy of the substances determination by standard methods and the completeness of the used one.

## Figures and Tables

**Figure 1 plants-11-00750-f001:**
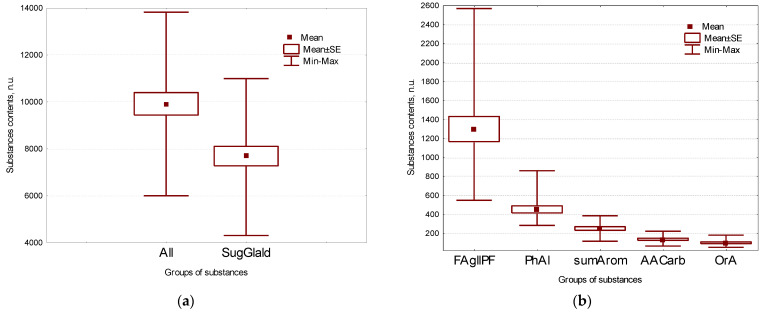
Box-whiskers plots of substances groups’ contents in seeds of all evaluated VIR flax genetic collection lines (in n.u.). (**a**) All substance, total sugars and glyceraldehyde; (**b**) total fatty acids and glycerol, polyhydric alcohols, secondary metabolites, amino acids and carbamide, organic acids and phosphate. Middle point is mean, Box is mean ± standard error, whiskers are min-max intervals.

**Figure 2 plants-11-00750-f002:**
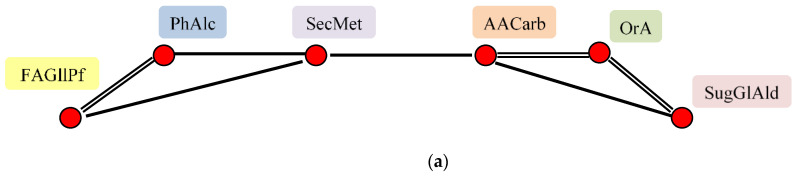
Correlation analysis of biologically active substances in flax seeds. (**a**) groups of substance; (**b**) sugars, glyceraldehyde; (**c**) fatty acids, glycerols, paraffin; (**d**) polyhydric alcohols; (**e**) secondary metabolites; (**f**) amino acids, carbamide; (**g**) organic acids. In (**b**–**d**) small red circles are separate substances, big circle is group of substances, related to each other 0.5 > r > 0.9, individual correlations are not shown, substances without circle are independent from the others. 
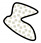
—All substances are related to each other 0.5 > r > 0.9, individual correlations are not shown. 
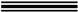
 r > 0.9, 

 0.9 > r > 0.7, 
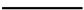
 0.7 > r > 0.5, 
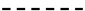
 r < 0.

**Figure 3 plants-11-00750-f003:**
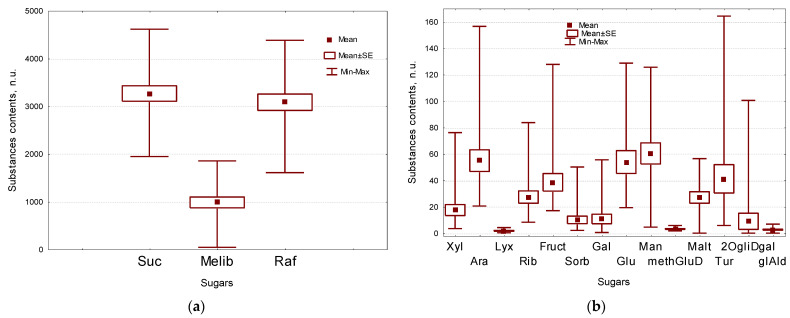
Box-whiskers plots of substances groups’ contents in seeds of all evaluated VIR flax genetic collection lines) (in n.u.). (**a**,**b**)–sugars; (**c**,**d**)–fatty acids, glycerols, paraffin; (**e**)–polyatomic alcohols, (**f**,**g**)–phytosterols, phenol-containing substances, heterocyclic aromatic substances; (**h**)–amino acids and carbamide; (**i**,**j**)–organic acids. Middle point is mean, Box is mean ± standard error, whiskers are min-max interval. Abbreviations see in Table 1.

**Figure 4 plants-11-00750-f004:**
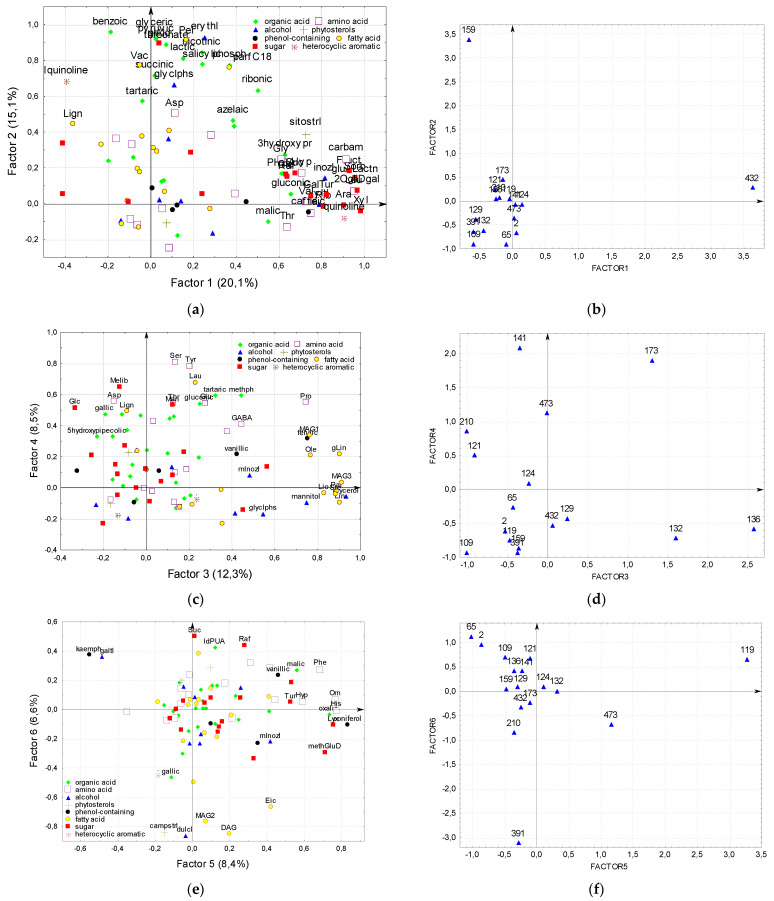
Factor analysis of flax metabolomic profile for 90 substances and 16 lines. (**a**) Factor 1 and Factor 2 system for substances; (**b**) Factor 1 and Factor 2 system for lines; (**c**) Factor 3 and Factor 4 system for substances; (**d**) Factor 3 and Factor 4 system for lines; (**e**) Factor 5 and Factor 6 system for substances; (**f**) Factor 5 and Factor 6 system for lines.

**Table 1 plants-11-00750-t001:** The amount of biologically active substances presented as average values (in n.u.) from all studied VIR flax genetic collection lines.

Abbreviation	Substance	Average ± Se	Abbreviation	Substance	Average ± Se
	**Sugars, glyceraldehyde**			**Secondary metabolites**	
Xyl	xylose	17.4 ± 4.2	campstrl	campesterol	38.2 ± 4.1
Ara	arabinose	54.8 ± 8.2	sitostrl	sitosterol	62.7 ± 3.2
Lyx	lyxose	1.72 ± 0.30	* **PhtSt** *	*total phytosterols*	*101 ± 5*
Rib	ribose	27.3 ± 4.6	coniferol	coniferol	15.9 ± 2.6
* **Pentose** *	*total pentoses*	*101 ± 16*	kaemph	kaempferol	26.2 ± 4.8
Fruct	fructose	38.4 ± 6.6	caffeic	caffeic acid	0.89 ± 0.28
Sorb	sorbose	9.97 ± 2.84	vanillic	vanillic acid	0.46 ± 0.09
Gal	galactose	10.7 ± 3.6	ferulic	ferulic acid	96.2 ± 19.5
Glc	glucose	53.7 ± 8.7	* **PhSb** *	*total phenolic substances*	*140 ± 19*
Man	mannose	60.3 ± 8.0	quinoline	quinoline	3.63 ± 0.86
methGluD	methyl glucose	3.06 ± 0.31	Iquinoline	isoquinoline	2.50 ± 0.53
* **Hexose** *	*total hexoses*	*176 ±* *20*	* **HCArom** *	*total heterocyclic aromaticsubstances*	*6.13 ± 0.74*
* **Msug** *	*total monosaccharides*	*277 ± 32*	* **SecMet** *	*total secondary metabolites*	*247 ± 19*
Suc	sucrose	3263 ± 162		**Amino acids, carbamide**	
Melib	melibiose	981 ± 113	Ala	alanine	14.2 ± 1.6
Malt	maltose	26.9 ± 4.3	Gly	glycine	10.4 ± 1.0
* **Dsug** *	*total disaccharides*	*4271 ± 229*	Pro	proline	20.0 ± 3.0
Raf	raffinose	3080 ± 171	Hyp	hydroxyproline	12.5 ± 1.4
Tur	turanoseglucopyranoside	41.0 ± 10.7	Val	valin	6.93 ± 0.62
* **Tsug** *	*total trisaccharides*	*3121 ± 179*	Leu	leucine	4.66 ± 1.74
* **Psug** *	*total polysaccharides*	*7392 ± 390*	Ser	serine	2.72 ± 0.45
2OgliDgal	2-O-glycerol-d-galactopyranoside	8.92 ± 6.12	Thr	threonine	5.99 ± 0.52
glAld	glyceraldehyde	2.59 ± 0.39	Asp	aspartic acid	21.0 ± 2.1
* **SugGlAld** *	*total sugars*	*7672 ± 413*	Orn	ornithine	0.90 ± 0.17
**Fatty acids, glicerol, paraffin**			Glu	glutamic acid	19.6 ± 3.7
pel	Pelargonic acid	1.00 ± 0.13	GABA	gamma-aminobutyric acid	0.49 ± 0.16
lau	lauric acid	0.29 ± 0.05	Phe	phenylalanine	9.83 ± 1.70
pal	palmitic acid	94.1 ± 9.5	His	histidine	1.12 ± 0.25
lio	linoleic acid	296 ± 27	Tyr	tyrosine	2.55 ± 0.46
ole	oleic acid	339 ± 37	carbam	carbamide	0.63 ± 0.13
lin	alpha-linolenic acid	349 ± 51	* **AACarb** *	*total amino acids and carbamid*	*134 ± 11*
vac	vaccenic acid	24.6 ± 3.5	**Organic acidsand phosphate**		
ste	stearic acid	51.1 ± 6.8	lactic	lactic acid	2.93 ± 0.25
eic	eicosanoic acid	7.79 ± 0.70	pyruvic	pyruvic acid	0.71 ± 0.14
glin	gamma-linolenic acid	40.6 ± 9.6	methylmalonic	methylmalonic acid	0.76 ± 0.21
lign	lignoceric acid	7.09 ± 1.19	oxalic	oxalic acid	3.48 ± 0.22
* **FA** *	*total fatty acids*	*1210 ± 128*	3hydroxypr	3-hydroxypropionic acid	0.98 ± 0.08
MAG1	monoacylglycerol 16:0	8.39 ± 1.36	phosph	phosphate	40.1 ± 3.7
MAG2	monoacylglycerol 18:2	35.5 ± 8.1	methph	methyl phosphate	1.00 ± 0.06
MAG3	monoacylglycerol 18:0	6.55 ± 0.99	benzoic	benzoic acid	0.40 ± 0.08
DAG	diacylglycerol	32.9 ± 3.2	PHBA	4-Hydroxybenzoic acid	1.14 ± 0.40
*Gll*	*total glycerols*	*83.3 ± 11.5*	nicotinic	nicotinic acid	0.22 ± 0.03
parC18	paraffin wax C18	1.65 ± 0.23	succinic	succinic acid	1.63 ± 0.29
* **FAGllPf** *	*total fatty acids and glycerols*	*1295 ± 132*	glyceric	glyceric acid	0.62 ± 0.08
	**Polyhydric alcohols**		5hydroxypipecolic	5-hydroxypipecolic acid	0.43 ± 0.05
erythl	erythritol	26.8 ± 1.6	malic	malic acid	10.2 ± 1.1
glycerol	glycerol	141 ± 30	Salicylic	salicylic acid	0.61 ± 0.10
glyclphs	glycerol-phosphate	16.4 ± 2.0	Threonate	threonic acid	0.89 ± 0.09
xyltl	xylitol	19.2 ± 4.6	Tartaric	tartaric acid	3.53 ± 0.52
mannitol	mannitol	12.2 ± 5.5	azelaic	azelaic acid	8.73 ± 1.01
dulcl	dulcitol	5.34 ± 0.77	ribonic	ribonic acid	2.13 ± 0.26
mInosl	myo-Inositol	27.3 ± 2.0	gluALactn	gluconic acid 1,5-lactone	10.8 ± 2.9
Inosl	inositol	132 ± 6	gallic	gallic acid	2.02 ± 0.32
galtl	galactinol	67.3 ± 10.8	gluconic	gluconic acid	1.16 ± 0.16
* **PhAlc** *	*total polyhydric alcohols*	*447 ± 37*	IdPUA	L-iduronic acid	1.01 ± 0.26
			* **OrA** *	*total organic acids*	*95.4 ± 8.7*
			* **All** *	*Total*	*9899 ± 480*

**Table 2 plants-11-00750-t002:** Comparison of the substances content in flax lines differing in morphological characteristics according to the criteria of U-Mann-Whitney and *t*-Student. *—differences between alternative groups by the analyzed parameter are significant at *p* < 0.05.

Substance	Morphological Character Presence						Test			
	Yes			No			U-Mann-Whitney		*t*-Student	
	*n*	Mean ± Se	Rank Sum	*n*	Mean ± Se	Rank Sum	Z Adjusted	*p*-Level *	*t*-Value	*p*-Level *
**Brown seeds**										
**Glu**	6	8.2 ± 3.8	28	10	26.5 ± 4.3	109	2.57	0.01	2.91	0.01
**Yellow seeds**										
**Iquinoline**	6	4.6 ± 0.8	80	10	1.2 ± 0.3	56	−3.15	0.002	4.77	0.0003
**White petals**										
**3hydroxypr**	4	0.66 ± 0.09	15	12	1.09 ± 0.05	121	2.30	0.02	2.71	0.02
**phosph**	4	26 ± 2	14	12	45 ± 4	122	2.43	0.02	2.50	0.03
**ribonic**	4	1.10 ± 0.21	13	12	2.48 ± 0.28	123	2.55	0.01	2.71	0.02
**OrA**	4	61 ± 6	14	12	107 ± 9	122	2.43	0.02	2.66	0.02
**Asp**	4	14 ± 2	16	12	24 ± 2	120	2.18	0.03	2.38	0.03
**sitostrl**	4	47 ± 4	10	12	68 ± 3	126	2.91	0.004	3.84	0.002
**Raf**	4	2312 ± 256	13	12	3336 ± 154	123	2.55	0.01	3.36	0.005

**Table 3 plants-11-00750-t003:** Characters of VIR genetic collection lines.

Line	Pedigree	Genes	Seeds Color	Characters
gc-2	l-1 from k-48, Altgauzen breeding, Russia		red-brown	wild type, earliness, high iodine number of oil
gc-65	l-3 from k-3178, local Russia, Tver region	*ora1*	speckled	orange anthers, earliness
gc-432	l-5 from k-4043, Deep Pink, The Netherlands	*pf-d*	black	pink petals, earliness
gc-109	l-3-2 from k-6099, Macovi M.A.G., Argentina	*wf1*	red-brown	white petals, early flowering
gc-119	l-2-3 from k-6210, NP (RR) 38, India	*dlb3-e, ora3*	red-brown	light blue petals, orange anthers, a lot of **ole** and few of **lin** in oil
gc-121	l-1-1 from k-6272, L.Dominion, N. Ireland	*sfc* *1, rs* *1*	light-yellow-brown	Violet petals, high yield
gc-124	l-1 from k-6284, Stormont Motley, N. Ireland	*f^e^*	spotted	earliness, dilution of anthocyanin pigmentation in all plant
gc-129	l-2 from k-6392, Bolley Golden, USA	*pf-ad yspf1*	yellow	pink petals, high iodine number of oil
gc-132	l-1 from k-6608, Currong, Australia	*sfbs1*	red-brown	no anthocyanins in hypocotyl and flower, white deformated petals, rust resistance, a lot of arabinoxilans in mucilage
gc-136	l-1 from k-6634, Mermilloid, Czech	*s1*	yellow	no anthocyanins in plant, white deformated petals, a lot of **ste** and **lin** in oil, high iodine number of oil, a lot of arabinoxilans, **Glc**, low amount of pectins in mucilage
gc-141	l-1 from k-6815, K-6, Russia	*pf1*	dark yellow-brown	pink petals
gc-159	l-1-1 from k-7659, Bionda, Germany	*YSED* *1*	yellow	a lot of arabinoxilans in mucilage, low amount of pectins and **Gal** in mucilage
gc-173	l-1 from i-548145, 48254, Ottawa 2152, Germany	*sgh1, sfc3-2, ysed2*	dark-yellow	violet petals, a lot of **ste** and **lio** in oil, a lot of arabinoxilans in mucilage
Γk-210	l-1 Иɜ i-588294, B-125, Lithuania	*ygp1*	red-brown	light blue petals, yellow-green color of the plant, a lot of arabinoxilans in mucilage
gc-391	l-1-2 from i-601679, Eyre, Australia	*sfbs1, YSED1, lufad3a, lufad3b*	yellow	no anthocyanins in the plant, white deformated petals, low content of **lin**, a lot of arabinoxilans in mucilage
gc-473	l-1 from i-606307, B-200, Lithuania		red-brown	yellow-green color of the plant

## Data Availability

All data is presented in the article.

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
