# Peer review of "Features of Profiles of Biologically Active Compounds of Primary and Secondary Metabolism of Lines from VIR Flax Genetic Collection, Contrasting in Size and Color of Seeds"

_plants, 2022, doi:10.3390/plants11060750_

Round 1
Reviewer 1 Report
Porokhovinova et al. described the seed metabolic profile of 16 lines of flax. The authors used factor analysis as the main way to examine the metabolic data and drive conclusions. I think factor analysis is not properly used in this study. Factor analysis is generally used to identify the relationship between variables (i.e. metabolites). However, the authors used the factor analysis to separate/classify samples (i.e. flax lines). The authors should consider the more-accepted Partial Least-Squares Discriminant Analysis for feature selection and classification. In addition, the writing of this paper has not been significantly improved compared to the last submission. The results and discussion are too descriptive, lacking hypothesis and biological significance.
- There has been no improvement on the introduction at all in this resubmission. As pointed out earlier, the introduction is extremely hard to read, with no flow of logic in 16 paragraphs. Starting with metabolomics may not be the best option. The authors should focus on their biological questions and then explain why metabolomics would be good to solve the problem. The introduction needs to be rewritten to have clear hypotheses and well-structured background information for the proposed research.
- However, several points were not addressed. The ions used for quantification and retention times should be provided in the supplementary information.
- Were calibration curves constructed for each analyte using external standards? If not, only relative abundance can be reported. Please don’t report the analyte concentration in ppm if the response factor of the standards for each analyte is not available.
- The entire results are still lengthy and not informative. Figure 3 is essentially the same data presented in Table 1. The correlation analysis in Figure 2 seems redundant. Isn’t it clear that compounds of the class would group? What new biological insights/hypotheses are provided from the correlation analysis?
- The added discussion only describes the comparison between the yellow and brown seed genotypes. There is not much to learn from the genotypic diversity within the 16 lines.
- No supplemental material is provided.
Author Response
Thank you very much for your valuable review of our article, which considerably helped us to improve it.
Unfortunately, we did not receive your comments after the first submission of the article and could not take them into account. Here are the answers for your comments.
Porokhovinova et al. described the seed metabolic profile of 16 lines of flax. The authors used factor analysis as the main way to examine the metabolic data and drive conclusions. I think factor analysis is not properly used in this study. Factor analysis is generally used to identify the relationship between variables (i.e. metabolites). However, the authors used the factor analysis to separate/classify samples (i.e. flax lines). The authors should consider the more-accepted Partial Least-Squares Discriminant Analysis for feature selection and classification. In addition, the writing of this paper has not been significantly improved compared to the last submission. The results and discussion are too descriptive, lacking hypothesis and biological significance.
Factor analysis has become a classic tool for processing of large packages of data. It identifies the general variability and groups the characters according to their variability, and then, based on the values of the characters that have the largest factor loads for each of the objects, groups the objects (in our case, flax lines) according to the factors obtained. So., in our experiment 6 factors were formed on the bases of 90 substances. Each factor is responsible for a certain group of substances and distributes all 16 lines based on them. And as can be seen from the initial data, it does this very accurately (see TableS 1 and fig 4), i.e. instead of a 16 x 90 matrix, a 16 x 6 matrix was formed.
This method has been known since the beginning of the 20th century and was first actively used in psychology, but now it is widely used in biodiversity research and has proven itself from the best side.
Unfortunately, discriminant analysis is not applicable to the analysis of a large number of characters, since only 3 - 5 substances will actually be used in the construction of the discriminant function and they will separate the most dissimilar genotype from the rest ones. This method works well for evaluation of several substances in many samples.But in our experiment we evaluated oppositely many substances and few genotypes.The same can be said about regression analysis, it is well applicable in cases when few substances are evaluated, but many repetitions of one experiment are done. We use all three methods in our research. Discriminant analysis clearly separates allied hybrids from a single crossing (allied genetic material) into classes according to 3-5 traits. Regression analysis is effective when there are a paired number of characteres.
The conclusions may seem to be vague, because the main conclusion is a description of flax intraspecific diversity by 90 characters, and it is impossible to mention all of them.
Our goal was to characterize the diversity of flax lines by 90 characters, primarily for researchers of plant biodiversity and breeders, and not to draw deep biochemical conclusions. Especially since we had practically no information about other investigations of flax metabolomics profiling to compare with our results. So the journal Plants, not Molecules, was chosen for the publication.
Our aim was also to promote a new method for the evaluation of a wide range of plants.
We have sufficiently characterized the entire matrix of 90 substances. Only methylmalonic and 5hydroxypipecolic acids, which were the only ones associated with the 7th factor,were not fully characterized. Factor analysis was carried out many times and the most optimal variant was chosen. Here is the "rocky scree":
- There has been no improvement on the introduction at all in this resubmission. As pointed out earlier, the introduction is extremely hard to read, with no flow of logic in 16 paragraphs. Starting with metabolomics may not be the best option. The authors should focus on their biological questions and then explain why metabolomics would be good to solve the problem. The introduction needs to be rewritten to have clear hypotheses and well-structured background information for the proposed research.
Metabolomic analysis in plants is comparatively a new method which has been used for a narrow assortment of objects. It is very different from the classical biochemical analysis. Our work on flax is pioneer. The main idea was to test this method for flax intraspecific diversity investigation. Therefore, we started the article with a discussion of the metabolome applicability as a method for the analyses of all groups of substances. 16 genetic collection lines were chosen for analysis, having different ecological and geographical origin, different seed color, fatty acids composition, chlorophyll and anthocyanin color. I.e., unlike similar works on other crops, not 2-3 groups of genotypes were analyzed, but 16, i.e. we have point wise covered almost all the variability known from flax.
- However, several points were not addressed. The ions used for quantification and retention times should be provided in the supplementary information.
The supplement’s table S1 is completed with retention index (RI) for all identified substances.
- Were calibration curves constructed for each analyte using external standards? If not, only relative abundance can be reported. Please don’t report the analyte concentration in ppm if the response factor of the standards for each analyte is not available.
L 689-698 – Since we have no calibration curves for all identified compounds, we determine the relative content of the components using the internal standard recalculation. Considering your righteous remark, we remove “mcg/g of dry mattert” and “ppm” from the text of the manuscript, and replace with nominal units (n.u.), not tied to specific units of measurement, for resalt’s comparition without a complete recalculation.
- The entire results are still lengthy and not informative. Figure 3 is essentially the same data presented in Table 1. The correlation analysis in Figure 2 seems redundant. Isn’t it clear that compounds of the class would group? What new biological insights/hypotheses are provided from the correlation analysis?
Figure 3 is similar to Table 3 only with average values and standard error. The main thing in Figure 3 is the variability interval, which can overlap the average value by 2-3 times. The standard error does not show this. Table 1 is necessary, since it gives a quantitative presentation of each character, and Figure 3 graphically expresses the scope of variability in the content of each of the substances and their ratio.
Correlation analyses showed that not all characters correlated with each other. Only substances which straightly depend on each other formed groups. For example, Xul and Ara are closely related, as they are formed as a result of the dissociation of arabinoxylans of the seed coat mucilage and are extracted in all lines in the same proportions. Oppositely, the amount of GlAld being completely independent of other identified substances’ concentrations, is a product of hexoses’ degradation and at the same time it is a precursor of all other sugars, and depends more on hardness of methanolis, then on the genotype.
Close correlations between the content of various substances may appear due to (a) their sequential biosynthesis or extraction, (b) their formation in the lines in a proportional amount. In case of substances independence from each others (without significant correlations), it could be hypothesed: (a) substances are extructed from different tissues, (b) independent biosynthesis, (c) substances are synthesised in a differente amounts in different genotypes.
The main conclusion from the biological and breeding points of view is that strong correlation between substances contents opens the possibility of simoltanious selection of veluable genotypes by detection of only one representative of th groupe. For example, the increase of treonin contents will also increase valin, serin, leucine, glycine and urea. Also, lines with high amount of threonine hihgly likely will form more listed aminoacids. Oppositely, the increase of kaempferol or campesterol whould not influence the concentration of other substances.
These results can be used in breeding and analyses of biodivercity.
- The added discussion only describes the comparison between the yellow and brown seed genotypes. There is not much to learn from the genotypic diversity within the 16 lines.
We have carried out a pioneer work. According to the data published before, the maximum number of samples in the metabolomic analysis was only 6, and the analysis had a much worse resolution. In our experiment the metabolomic analysis was done in semi-automatic mode, each of the 90 (and even more) peaks for 16 lines has been checked against the standard. 16 lines were selected from the collection which consists of more than 6000 accessions in total. Chosen lines represent a wide range of different characters diversity.
- No supplemental material is provided.
Supplementary materials were uploaded together with the article.

Reviewer 2 Report
I have no comments at this time.
Author Response
Thank you very much for your valuable review of our article, which considerably helped us to improve it.
Reviewer 3 Report
Dear Authors and Editor,
I read the revised manuscript, as well as the author response file. I'm really satisfied with the corrections. The Results and Discussion and Materials and Methods have been improved.
Please write correct abbreviation of Inositol in Figure 4.
Recommendation: Accept Submission
Author Response

(The authors gave the same response as above.)

Reviewer 4 Report
Dear Authors
The manuscript entitled "Features of profiles of biologically active compounds of primary and secondary metabolism of lines from VIR flax genetic collection, contrasting in size and color of seeds" aimed to identify differences in the metabolomic profiles of flax lines contrasting in color and size of seeds and suggested metabolomic profiles is promising for a comprehensive assessment of the VIR flax genetic collection.
The manuscript has been already improved significantly, has been planned and presented very well. Conclusion can be reorganised in shorter and clearer way to convey the key message of the study.
I do not have any further query.
Thank you
Author Response

(The authors gave the same response as above.)

Round 2
Reviewer 1 Report
The authors have responded prior comments.